# Control of Macrophage Inflammation by P2Y Purinergic Receptors

**DOI:** 10.3390/cells10051098

**Published:** 2021-05-04

**Authors:** Dominik Klaver, Martin Thurnher

**Affiliations:** Immunotherapy Unit, Department of Urology, Medical University of Innsbruck, 6020 Innsbruck, Austria; dominik.klaver@i-med.ac.at

**Keywords:** P2Y receptor, G protein, P2Y_11_, P2Y_14_, ATP, UDP-glucose, macrophage, inflammation

## Abstract

Macrophages comprise a phenotypically and functionally diverse group of hematopoietic cells. Versatile macrophage subsets engage to ensure maintenance of tissue integrity. To perform tissue stress surveillance, macrophages express many different stress-sensing receptors, including purinergic P2X and P2Y receptors that respond to extracellular nucleotides and their sugar derivatives. Activation of G protein-coupled P2Y receptors can be both pro- and anti-inflammatory. Current examples include the observation that P2Y_14_ receptor promotes STAT1-mediated inflammation in pro-inflammatory M1 macrophages as well as the demonstration that P2Y_11_ receptor suppresses the secretion of tumor necrosis factor (TNF)-α and concomitantly promotes the release of soluble TNF receptors from anti-inflammatory M2 macrophages. Here, we review macrophage regulation by P2Y purinergic receptors, both in physiological and disease-associated inflammation. Therapeutic targeting of anti-inflammatory P2Y receptor signaling is desirable to attenuate excessive inflammation in infectious diseases such as COVID-19. Conversely, anti-inflammatory P2Y receptor signaling must be suppressed during cancer therapy to preserve its efficacy.

## 1. Introduction

Elie Metchnikoff (1845–1916) not only first described macrophages but also recognized that a fundamental characteristic of these cells is to strive for balance [1]. The surveillance function of macrophages depends on the ability to sense even delicate changes in the tissue microenvironment [2]. For this purpose, macrophages are endowed with a variety of receptors that respond to molecular patterns arising in intra- and extracellular spaces during cellular stress [3]. Receptor-mediated signal transduction causes genetic reprogramming and activates macrophage functions that in turn contribute to the restoration and maintenance of cellular integrity and tissue homeostasis.

Application of the Th1/Th2 paradigm [4] to macrophage activation and differentiation resulted in the concept of M1/M2 macrophage polarization, which provided a useful framework for macrophage research, although it does not sufficiently reflect macrophage plasticity in vivo [5]. M1 macrophages (also known as classically activated macrophages) are pro-inflammatory anti-microbial macrophages. M1 macrophages mediate delayed type hypersensitivity (DTH) and constitute the first line of defense against intracellular pathogens. In addition, M1 macrophages support Th1 polarization of CD4^+^ T helper cells by providing IL-12. In contrast, M2 macrophages (also known as alternatively activated macrophages) are required to regulate or even turn off the inflammatory response. The M2 subset can be further subdivided into M2a and M2b macrophages, which drive Th2 responses, as well as M2c macrophages involved in immune deactivation, scavenging, and tissue remodeling (Figure 1) [6]. Finally, the tumor microenvironment may promote the differentiation of a distinct phenotype that has been referred to as M2d [7] (Figure 1).

Upon infection, macrophages recognize pathogen-associated molecular patterns (PAMPs) [3]. In the absence of pathogens, damage-associated molecular patterns (DAMPs), which are derived from stressed host cells, may induce a sterile form of inflammation. PAMPs and DAMPs are recognized by pattern recognition receptors, including Toll-like receptors (TLRs), cytoplasmic NOD-like receptors (NLRs), intracellular retinoic acid-inducible gene-I)-like receptors (RLR), transmembrane C-type lectin receptors, and AIM2 (absent in melanoma 2)-like receptors (ALRs). In addition, macrophages respond to PAMP- and DAMP-induced cytokines and lipids as well as to immune complexes, the products of antibodies binding to antigens. One of the best characterized PAMP is lipopolysaccharide (LPS), which is recognized by TLR4. In collaboration with interferon-γ, LPS promotes classical macrophage activation (M1) [6]. In contrast, IL-4 and IL-13, which are produced during Th2-polarized responses, mediate alternative macrophage activation (M2) to control helminth infection but also promote pathophysiological conditions such as allergies (Figure 1).

The purinergic P2X and P2Y receptors represent yet another group of receptors capable of sensing cell and tissue perturbation [8,9]. While P2X receptors are ligand-gated ion channels [10], P2Y receptors are class A G protein-coupled receptors (GPCRs) [11] and belong to the δ group of rhodopsin-like GPCRs. The P2Y receptor family comprises eight subtypes: five G_q_-coupled, P2Y_1_-like receptors (P2Y_1_, P2Y_2_, P2Y_4_, P2Y_6_ and P2Y_11_) and three G_i_-coupled, P2Y_12_-like receptors (P2Y_12_, P2Y_13_ and P2Y_14_) (Table 1).

Like other GPCRs, P2Y receptors consist of seven hydrophobic transmembrane segments connected by three extracellular loops (ECL) and three intracellular loops, with an extracellular N-terminus and an intracellular C-terminus. An ECL serves to bind the receptor ligand(s), while intracellular regions mediate G protein activation and participate in P2Y receptor regulation. P2Y receptors can form both homodimers and heterodimers to further increase the biochemical and pharmacological spectrum of P2YRs.

P2Y receptors respond to nucleotides (ATP, ADP, UTP and UDP) as well as to nucleotide sugars such as UDP-glucose. Basal release of these molecules is weak. However, in response to various forms of stress, the level of nucleotides and nucleotide sugars in the extracellular space may rise dramatically either via controlled export (pannexin-1 hemichannels, secretory vesicles) or via lytic release [12,13] leading to the activation of P2Y receptors [14]. Strength and duration of P2Y receptor responses are controlled by ectoenzymes such as the the ecto-ATPase CD39 that mediate nucleotide degradation [15].

P2Y receptor signaling cascades are mainly initiated by heterotrimeric G proteins (G_αβγ_) bound to the GPCR at the inner side of the plasma membrane (C-terminus). Upon P2Y receptor activation, heterotrimeric G proteins dissociate into G_α_ subunits and G_βγ_ complexes, which activate or regulate downstream effector proteins. However, P2Y receptors may also recruit β-arrestins leading to the activation of alternative signaling pathways, including mitogen-activated protein kinase (MAPK) activation [16,17]. P2Y receptors thus control a variety of cellular processes related to inflammatory and immune responses, including cell differentiation, adhesion, migration, phagocytosis, and secretion.

In this review, we summarize current knowledge of P2Y receptor expression and function in monocytes and macrophages. In addition, we discuss the role of P2Y receptors in macrophage-regulated physiological and pathogenic inflammation.

## 2. P2Y_1_ Receptor

P2Y_1_ receptors are relatively insensitive to adenosine triphosphate (ATP), but are strongly activated by adenosine diphosphate (ADP) [11], indicating that ADP is the preferred natural agonist for the G_q_-coupled P2Y_1_ receptor. ATP may act as a relatively weak partial agonist or as an antagonist at the P2Y_1_ receptor [18]. In the canonical metabotropic pathway, coupling of P2Y_1_ receptor to G_q_ in response to ADP causes activation of phospholipase Cß (PLCß) resulting in the hydrolysis of phosphatidylinositol 4,5-bisphosphate [PtdIns(4,5)P_2_ or PIP_2_] to produce the second messenger Ins(1,4,5)P3 (IP_3_), which mobilizes intracellular Ca^2+^ via IP_3_ receptors.

Knockout mice lacking P2Y_1_ receptor have no overt phenotype [19]. However, the lack of P2Y_1_ expression was found to increase bleeding time and to protect from collagen- and ADP-induced thromboembolism [20]. Today, it is well established that the P2Y_1_ receptor is a crucial factor in the initiation of platelet activation induced by ADP or collagen [21].

P2Y_1_ receptor-encoding mRNA could be detected in human monocytes [22,23] as well as in human monocyte-derived dendritic cells generated with GM-CSF plus IL-4, and in human monocyte-derived macrophages generated on IgG-coated surfaces in the presence of GM-CSF [24]. During differentiation of human macrophages from monocytes, P2Y_1_ receptor mRNA was found to be significantly more abundant in GM-CSF differentiated M1-like pro-inflammatory macrophages compared to M-CSF differentiated M2-like anti-inflammatory macrophages [25]. In addition, human alveolar macrophages were also found to express P2Y_1_ receptor mRNA [26]. In mice and rats, P2Y_1_ receptor-encoding mRNA was detectable in peritoneal macrophages [27,28,29].

While P2Y_1_ receptor-encoding mRNA could readily be detected in monocytes and monocyte-derived cells, evidence of functional P2Y_1_ receptor expression is very limited. While bacterial infection has been shown to enhance ADP release and functional P2Y receptor expression in macrophages, P2Y_1_ receptor was found to be dispensable for ADP-mediated protection against bacterial infection [30]. The role of P2Y_1_ receptors in Ca^2+^ signaling has been explored in mouse peritoneal macrophages [28]. Application of 100 μM ADP, which was ATP-depleted with hexokinase, had no effect on intracellular calcium, whereas ATP produced the expected strong response, bolstering the argument against functional P2Y_1_ receptor expression.

The dinucleotide NAD^+^, which shares structural characteristics with ATP, is another key regulator of metabolic and inflammatory processes [31] and may likewise serve as a danger signal in the extracellular space. In human monocytes isolated by counterflow elutriation, extracellular NAD^+^ has been shown to increase intracellular Ca^2+^ concentration by internal calcium mobilization and calcium influx [32]. However, NAD^+^ is not an agonist at the P2Y_1_ receptor [11] and therefore more likely acted by stimulating basal P2Y_1_ receptor activity via calcium mobilization. Nevertheless, the NAD^+^-induced calcium response in activated monocytes could be inhibited partially by the P2Y_1_ receptor antagonist MRS2179, providing at least some evidence for functional P2Y_1_ receptor expression in monocytes.

Osteoclasts are multinucleated cells formed by the fusion of mononuclear progenitors of the monocyte/macrophage family [33] (Figure 1). Maturation of macrophages into osteoclasts requires the presence M-CSF and receptor for activation of nuclear factor kappa B (NF-κB) (RANK) ligand (RANKL). ADP has been shown to stimulate osteoclast bone resorption in vitro, in part through the P2Y_1_ receptor on osteoclasts [34]; however, P2Y_12_ has later been shown to be the primary ADP receptor of osteoclasts involved in bone resorption [35].

Taken together, current data do not support a critical role of the P2Y_1_ receptor monomer in monocyte and macrophage biology. P2Y_1_ may nevertheless exert effects in monocytes and macrophages by forming heterodimers with P2Y_11_ [36], which are more difficult to study. In addition, macrophage-mediated inflammation may differentially affect P2Y_1_ function in other cell types [37,38].

## 3. P2Y_2_ Receptor

The P2Y_2_ receptor is activated equally well by ATP and uridine triphosphate (UTP) [39,40]. Stimulation of the P2Y_2_ receptor with ATP or UTP causes G_q_-dependent activation of PLCß followed by intracellular Ca2^+^ mobilization via IP_3_ receptors and diacylglycerol (DAG)-induced protein kinase C (PKC) activation. The Ca^2+^ signal can be relayed and amplified by calmodulin (CaM) and calmodulin-dependent kinase (CaMK) leading to nitric oxide (NO) formation via nitric oxide synthase (NOS) and guanylyl cyclase (GC)-driven synthesis of cyclic GMP (cGMP), which in turn activates protein kinase G (PKG). Ca^2+^ and PKC cooperatively activate cytosolic phospholipase A2 (cPLA_2_), liberating arachidonic acid (AA) from membrane phospholipids, which is the precursor of prostaglandins (PGs) such as PGE2 [41]. P2Y_2_ receptor activation has also been shown to induce oscillations in cytosolic Ca^2+^ [42].

Like other GPCRs, P2Y_2_ receptor can transactivate growth factor receptors, including epidermal growth factor receptor (EGFR), platelet-derived growth factor receptor (PDGFR), and vascular endothelial growth factor receptor-2 (VEGFR-2) [43]. In recombinant 1321N1 astrocytoma cells, P2Y_2_ agonist induces direct interaction of Src with the SH3 binding sites in the P2Y_2_ receptor to facilitate Src activation, which then recruits the EGFR into a protein complex with the P2Y_2_ receptor and allows Src to phosphorylate and activate the EGFR [44]. P2Y_2_-mediated EGFR transactivation may also occur through activation of the metalloproteases ADAM10 and ADAM17, causing the cleavage and release of membrane-associated EGFR ligands, which activate EGFR [45,46].

By virtue of an RGD sequence within an extracellular motif, the P2Y_2_ receptor can also interact with α_V_β_3_ integrin and induce integrin-dependent activation of cell migration and pro-inflammatory responses. The RGD domain was required for P2Y_2_ receptor-mediated signal transduction, which was at least in part mediated by G_o_ proteins [47,48]. P2Y_2_ receptor-induced clustering of αv integrins was shown to be required for the activation of extracellular signal-regulated kinases 1 and 2 (ERK 1/2) [49].

P2Y_2_ receptor mRNA is expressed in human monocytes [22,23] and in monocyte-derived dendritic cells and macrophages [24]. In rodents, P2Y_2_ receptor expression has been detected in murine [26,28,50,51] and rat [27,52] alveolar and peritoneal macrophages as well as in murine and rat brain-resident macrophages (microglia) [27,52,53]. In addition, the P2Y_2_ receptor is expressed by osteoclasts [54].

The P2Y_2_ receptor expressed on human and mouse macrophages senses ATP and UTP released by apoptotic cells [55]. P2Y_2_ receptor-directed migration thus enables macrophages to find apoptotic cells for subsequent phagocytic clearance, which is prerequisite for tissue homeostasis. During immune responses to microbial infection, macrophages also eliminate bacteria through phagocytic clearance. In human macrophages, ATP-activated P2Y_2_ receptors may be involved in the killing of engulfed mycobacteria by promoting acidification of mycobacteria-containing phagosomes [56]. In the same study, human macrophages were found to co-express P2Y_11_ receptor that may be required to prevent ATP-induced macrophage death during ATP/P2Y_2_-mediated killing of mycobacteria.

P2Y_2_ receptor has also been implicated in the regulation of macrophage migration along gradients of the complement component C5a. Mouse peritoneal macrophages have been shown to sense the chemoattractant C5a, resulting in chemotaxis driven by ATP release and autocrine stimulation of P2Y_2_ and P2Y_12_ receptors [57].

ATP and UTP were shown to shape the pro-inflammatory response of macrophages via P2Y_2_ receptor signaling. Application of UTP and ATP (100 µM) during a 3-h macrophage priming enhanced IL-1β production stimulated by LPS-induced NLRP3 inflammasome activation in murine peritoneal macrophages [51]. Pretreatment of macrophages with the P2Y_2_ receptor inhibitor AR-C118925xx prevented the observed effects, and priming with nucleotides failed in macrophages isolated from mice lacking the P2Y_2_ receptor. Conversely, P2Y_2_ receptor agonistic nucleotides decreased the production of the proinflammatory cytokine TNF-α in response to LPS stimulation. Intriguingly, the priming effect of nucleotides was abolished at macrophage densities greater than 10^6^ cells/mL. Treatment of RAW264.7 murine macrophages with LPS for 4 h resulted in upregulation of P2Y_2_ receptor mRNA and thus resulted in enhanced HMGB1 secretion, COX-2 and iNOS expression as well as PGE2 and NO production in response to stimulation with UTP or ATP [58].

Nanoparticles (nano-sized inorganic metal oxides such as silica dioxide and titanium dioxide) have been shown to upregulate P2Y_2_ receptor expression and P2Y_2_ receptor signaling may participate in nanoparticle-induced NLRP3 inflammasome-dependent IL-1ß secretion [59]. P2Y_2_ receptors have been shown to control leishmanial infection. Infection of murine peritoneal macrophages with Leishmania amazonensis, the causative agent of cutaneous leishmaniasis, upregulated the expression of P2Y_2_ and P2Y_4_ receptor-encoding mRNAs [60]. The P2Y_2_ receptor agonist UTP strongly reduced parasite load. Similar to P2Y_2_ receptor-mediated killing of phagocytosed mycobacteria [56], uridine nucleotides also induced selective production of reactive oxygen species (ROS) and NO in infected macrophages, causing dose-dependent macrophage apoptosis. Of note, the survival-promoting P2Y_11_ receptor that prevents P2Y_2_-induced cell death in mycobacteria-containing human macrophages [56], is absent in mice [61]. P2Y_2_ is thus the sole G_q_-coupled receptor for UTP or ATP in mouse macrophages [28]. The authors also proposed that heteromerization of P2Y receptors on macrophages during Leishmania infection contributes to the observed nucleotide responses [60]. The same group reported crosstalk between P2Y_2_ and P2X_7_ receptors in the control of cutaneous leishmaniasis [62]. P2Y_2_ receptor signaling in response to relatively low levels of ATP (50–100 µM) reduced Leishmania amazonensis infection, an effect that could be abolished by P2Y_2_ receptor antagonist. UTP treatment increased ATP release from infected macrophages in a pannexin-1-dependent manner, which in turn activated P2X_7_ receptors, and thus amplified the anti-microbial capacity of infected macrophages. P2Y_2_ receptor also contributed to resistance of macrophages against infection with the protozoan parasite Toxoplasma gondii. P2Y_2_, P2Y_4_, and P2Y_6_ agonists strongly reduced parasite load, however not by inducing intracellular parasite killing but rather by promoting premature parasite egress from infected peritoneal mouse macrophages, resulting in reduced infectivity of the “thrown out” parasites [63].

In addition to its protective roles, P2Y_2_ receptor is also a known effector in a cardiopulmonary syndrome, which is induced by hantavirus and mediated by a dysregulated immune response [64]. Lung macrophages expressing P2Y_2_ receptor were found to be the primary virus reservoir and P2Y_2_ receptor mRNA was upregulated in cases of cardiopulmonary syndrome. Integrin-dependent P2Y_2_ receptor activation is thought to contribute to pathogenesis by initiating the coagulation cascade [64], stimulating vascular permeability and leukocyte homing to sites of infection.

Other examples of P2Y_2_ receptor involvement in disease pathogenesis include upregulation of P2Y_2_ receptor on macrophages during smoke-induced lung inflammation [50] and the increased influx of P2Y_2_ receptor-expressing macrophages in atherosclerosis [65].

Taken together, P2Y_2_ receptor is abundant in the mononuclear phagocytic system of humans and rodents. P2Y_2_ receptor is critically involved in tissue surveillance by for instance removing apoptotic cells and eliminating pathogens. Consistent with the adaptive functions of metabotropic P2Y receptors, P2Y_2_ receptors may have both pro- and anti-inflammatory effects.

## 4. P2Y_4_ Receptor

The P2Y_4_ receptor is activated by UTP, but not by nucleoside diphosphates [11]. In contrast to the P2Y_2_ receptor, ATP is not an agonist at the P2Y_4_ receptor. Instead, ATP can antagonize the UTP-induced responses at the P2Y_4_ receptor [66]. Like P2Y_1_ and P2Y_2_ receptors, the P2Y_4_ receptor couples mainly to G_q/11_ proteins. However, the P2Y_4_ receptor response could partially be inhibited by pertussis toxin, suggesting that the P2Y_4_ receptor also couples to G_i_ proteins [67]. P2Y_4_ receptors are expressed in heart, lung, and placenta. Studies with knockout mice revealed that cardiac endothelial cells lacking P2Y_4_ receptor have reduced proliferative, migratory, and secretory capacity, resulting in reduced postnatal cardiac development also due to impaired communication between endothelial cells and cardiomyocytes [68]. The P2Y_4_ receptor has also been implicated in ATP-induced chloride secretion in the gut [69].

P2Y_4_ receptor mRNA could be detected in human monocyte-derived dendritic cells generated with GM-CSF plus IL-4 at various stages of differentiation and activation [24]. In contrast, macrophages differentiated from highly purified human monocytes with recombinant M-CSF (5 ng/mL) did not express P2Y_4_ receptor mRNA [42], suggesting that resting M2 macrophages lack a P2Y_4_ receptor. In another study, macrophages were generated by plating peripheral blood mononuclear cells onto IgG coated bacteriological dishes and culturing the adherent fraction in the presence of GM-CSF (800 U/mL) [24]. These macrophages expressed P2Y_4_ receptor-encoding mRNA. It is unclear whether detection of P2Y_4_ receptor mRNA in the latter study is due to contamination of non-monocytic cells that may also be selected by the IgG-coating protocol or due to the use of GM-CSF, which may be a better stimulus than M-CSF [25].

Infection of murine peritoneal macrophages with Leishmania amazonensis, the causative agent of cutaneous leishmaniasis, has been shown to upregulate the expression of P2Y_4_ receptor-encoding mRNAs [60]. In addition, P2Y_4_ agonist was found to reduce parasite load [63].

In contrast to P2Y_2_ receptor, evidence of functional P2Y_4_ receptor in monocytes and macrophages remains limited. GM-CSF, which drives dendritic cell and M1 macrophage generation, may be a better driving force of P2Y_4_ receptor expression than M-CSF, which promotes M2 macrophage development. However, activation of M2 macrophages during infection may upregulate P2Y_4_ receptor expression and elicit P2Y_4_ receptor-induced anti-microbial effects.

## 5. P2Y_6_ Receptor

The G_q_-coupled P2Y_6_ receptor is preferentially activated by UDP [11,67]. UDP-mediated activation of P2Y_6_ receptor promotes the recruitment of inflammatory cells to sites of inflammation or infection by inducing the production of various chemokines including CCL2 (monocyte chemotactic protein 1, MCP-1), CXCL8 (IL-8) and CCL20 (macrophage inflammatory protein-3α, MIP-3α) [70,71,72,73,74].

Although the P2Y_6_ receptor clearly has pro-inflammatory potential, it may also have protective effects. P2Y_6_ receptor-deficient mice were more susceptible to inflammation in the dextran sodium sulfate (DSS) murine model of inflammatory bowel disease (IBD) [75]. The P2Y_6_ receptor appears to prevent the development of IBD at least in part via blockade of T helper 17 (Th17) cells.

P2Y_6_ receptor-encoding mRNA could be detected in human monocytes enriched by plastic adherence and in promonocytic U937 cells [22]. In human monocyte-derived macrophages, P2Y_6_ receptor mRNA was much more abundant in M-CSF-induced M2 macrophages than in GM-CSF-induced M1 macrophages [25]. Moreover, M-CSF was observed to stimulate P2Y_6_ receptor protein expression [76]. Human dendritic cells and macrophages generated with GM-CSF were also shown to express P2Y_6_ receptor mRNA [24]. In addition, P2Y_6_ receptor mRNA has been detected in human alveolar macrophages [26] and in mouse peritoneal macrophages. However, resting mouse macrophages showed weak responses to UDP (purified by hexokinase to remove ATP). Only upon activation did UDP responsiveness (i.e., P2Y_6_ receptor expression) increase [28]. In another mouse study, P2Y_6_ receptor mRNA expression was found to be low in bone marrow-derived macrophages, abundant in peritoneal macrophages, and particularly high in RAW264.7 cells [29].

P2Y_6_ receptor was found to be required for M-CSF induced human macrophage differentiation, since P2Y_6_ receptor knockdown prevented M-CSF induced macrophage differentiation from monocytes [76]. Autophagy is induced during M-CSF driven macrophage differentiation through activation of AMP-activated kinase (AMPK). P2Y_6_ receptor signaling serves to maintain the autophagic process or to mediate its re-induction to ensure successful macrophage differentiation [77].

UDP-mediated P2Y_6_ receptor activation stimulated CXCL8 release in human THP-1 monocytic cells [78]. In addition, LPS appears to induce CXCL8 production at least in part by autocrine P2Y_6_ receptor activation. UDP-mediated P2Y_6_ receptor activation also increased CCL20 release from monocytes through NF-kB and MAPK pathways (ERK1/2 and p38) [79]. Proinflammatory P2Y_6_ receptor may be involved in pathophysiological processes such as atherosclerosis by augmenting pro-inflammatory responses in macrophages [80]. Vascular inflammation and atherosclerosis are limited in mice lacking a P2Y_6_ receptor [65].

Conversely, proinflammatory P2Y_6_ receptors may be protective during allergen exposure. P2Y_6_ receptor signaling in murine alveolar macrophages has been shown to prevent type 2 (allergic) lung inflammation [81]. P2Y_6_ receptor promotes an innate type 1 response involving macrophage-derived IL-12 and NK cell-derived IFN-γ that prevent pathologic type 2 immune responses to respiratory allergens. Likewise, UDP/P2Y_6_ receptor signaling has been shown to be protective during vesicular stomatitis virus (VSV) infection [82]. UDP released from VSV-infected cells activates P2Y₆ receptor on other cells including macrophages and promotes anti-viral immunity via interferon-ß secretion. As mentioned before, in Toxoplasma gondii infection, UDP-mediated P2Y_6_ receptor activation can strongly reduce parasite load in infected macrophages [63].

The P2Y_6_ receptor is also upregulated in rat brain when neurons are damaged. Stress-dependent release of endogenous UDP triggers phagocytosis by microglia, which then engage in the clearance of dead cells or dangerous debris [83]. P2Y₆ receptor may also participate in chemokine receptor signaling in macrophages. P2Y₆ receptor is activated when CCL2 binds to its receptor CCR2 [84], a receptor that mediates monocyte egress from the circulation and infiltration of inflamed tissues. In this context, P2Y₆ receptor-mediated calcium mobilization serves to support chemokine receptor signaling.

Uric acid (urate) released from injured or infected cells serves as a danger signal. Poorly soluble urate precipitates and forms monosodium urate (MSU) crystals. Uptake of MSU crystals by macrophages stimulates inflammatory cytokine production in an inflammasome-dependent manner via caspase-1 mediated maturation and secretion of bioactive IL-1 [85]. P2Y₆ receptor has been implicated in MSU-driven inflammation [72], since antagonism or knockdown significantly inhibited the IL-1 driven inflammatory response to MSU crystal exposure.

UDP-induced secretion of chemokines CCL2 and CCL3 (macrophage inflammatory protein 1α, MIP-1α) from microglia has also been reported [86]. The response could be attributed to P2Y_6_ receptors by specific receptor antagonism and siRNA-mediated knockdown. Consistent with G_q_-coupling of P2Y_6_ receptor, inhibition of PLC and calcium increase reduced UDP-induced chemokine expression. The calcium-activated transcription factors NFATc1 and NFATc2 (Nuclear Factor of Activated T cells) were also shown to be involved. Moreover, P2Y_6_ receptor has been shown to support TLR2-induced CXCL8 release from human monocytes [71].

P2Y_6_ receptors also regulate the function of human dendritic cells [87]. UDP stimulated chemotaxis of immature dendritic cells and the release of the chemokine CXCL8 in mature dendritic cells. Finally, UDP-activated P2Y_6_ receptors have been shown to stimulate the formation of osteoclasts from precursor cells and concomitantly enhance the resorptive activity of mature osteoclasts [88]. In addition, P2Y_6_ receptors activated NF-kB to increase osteoclast survival [89].

In summary, P2Y_6_ receptors are upregulated during macrophage differentiation and appear to be particularly required for M-CSF driven macrophage differentiation. P2Y_6_ receptor activity may depend on the stage of macrophage activation. P2Y_6_ receptors can contribute to the restoration of homeostasis by clearing dead cells or dangerous debris. As a pro-inflammatory receptor, P2Y_6_ is protective against pathogens but may also promote inflammatory disease or osteoporosis. By stimulating Th1-type inflammation, P2Y_6_ receptors may prevent allergic disease (Th2-type) or ameliorate autoimmune disease (Th17-type).

## 6. P2Y_11_ Receptor

The G_q_- and G_s_-coupled P2Y_11_ receptor is activated by ATP and its slowly hydrolyzed analog ATPγS [11,67]. Somewhat surprisingly, NF546 can also activate P2Y_11_ receptors, despite the fact that the compound is derived from the non-selective P2 purinergic antagonist suramin [90]. The suramin analog NF340 is currently the most useful antagonist at the P2Y_11_ receptor [61,90,91]. P2Y_11_ receptors can activate canonical G_q_-mediated signaling through PLCβ. However, the P2Y_11_ receptor is the only P2Y family member which can also raise intracellular levels of cyclic AMP (cAMP) by G_s_-mediated activation of AC. The dinucleotide NAD^+^, which-like ATP-serves as a key regulator of metabolic and inflammatory processes [31], has also been considered an agonist at the P2Y_11_ receptor expressed by human granulocytes [92], but this could not be confirmed in human macrophages [93]. Extracellular NAD^+^ can permeate the plasma membrane and replenish intracellular NAD^+^ pools. By serving as a metabolite rather than a receptor agonist NAD^+^ may raise intracellular cAMP and promote influx of extracellular Ca^2+^ [94], conditions that may facilitate constitutive P2Y_11_ receptor activity.

P2Y_11_ receptor expression is often associated with immune cells. P2Y_11_ receptors have recently been shown to participate in the process of T cell migration by regulating mitochondrial metabolism [95]. In addition to T cells, which are components of the adaptive immune system, P2Y_11_ receptor expression has repeatedly been reported to occur in innate immune cells [25,61,90,91,93,96]. Human monocyte-derived dendritic cells and human monocyte-derived macrophages have been shown to express P2Y_11_ receptor mRNA [24]. Functional P2Y_11_ expression supported by pharmacological data has first been postulated for monocyte-derived dendritic cells [90,96]. However, in later studies, P2Y_11_ expression has been demonstrated to strongly increase during macrophage differentiation [25,93]. P2Y_11_ receptor mRNA was found to be significantly more abundant in M-CSF differentiated M2-like macrophages compared to GM-CSF differentiated M1-like macrophages [25]. In a study of human macrophages generated with M-CSF from highly purified monocytes, P2Y_11_ mRNA was much more abundant than other P2Y receptor mRNAs [42]. Our recent finding that P2Y_11_ receptor protein is upregulated during M2 macrophage differentiation [93] represented the first example of P2Y_11_ receptor regulation at the protein level. In addition, we found that IL-10, which drives differentiation of highly anti-inflammatory M2c macrophages (Figure 1), further enhanced P2Y_11_ receptor protein expression [93], suggesting that P2Y_11_ receptor contributes to the anti-inflammatory potential of M2 (M2c) macrophages. Consistent with earlier findings in dendritic cells [90,96], P2Y_11_ receptor activation in M2 macrophages induced CXCL8 secretion. We extended these studies and performed the first transcriptomic and secretomic analyses of P2Y_11_ receptor activation using an ectopic expression system [97]. The transcriptional network analyses revealed the signature of a strong but transient interleukin-1 receptor (IL-1R)/toll-like receptor (TLR) response, including activation of the IL-1R/TLR target genes IL6 and IL8. Secretome profiling of P2Y_11_ receptor activation confirmed the production of IL-6 and CXCL8, and additionally identified soluble tumor necrosis factor receptor 1 and 2 (sTNFR1 and sTNFR2) as P2Y_11_ targets. While recombinant astrocytoma cells released sTNFR1 but not sTNFR2, M2 macrophages predominantly released sTNFR2 in response to P2Y_11_ activation. Non-specific inhibition of phosphodiesterases (PDEs) with IBMX or specific inhibition of PDE4 with rolipram, which raises intracellular levels of cAMP [98], significantly enhanced P2Y_11_-driven sTNFR2 release. IBMX is also known to inhibit LPS-induced TNF-α production in macrophages [99]. Accordingly, P2Y_11_ activation with ATPγS effectively inhibited TNF-α production in LPS-activated macrophages, an effect that could be completely prevented by the P2Y_11_ antagonist NF340. Our data thus indicate that P2Y_11_ signaling via a G_s_-AC-cAMP axis blocks TNF-α secretion and promotes the release of sTNFR2 in human M2 macrophages (Figure 2), facilitating the resolution of inflammation [97].

P2Y_11_ is a rather unconventional member of the P2Y family of G protein-coupled receptors. It is the only P2Y receptor that couples to both G_q_ and G_s_ proteins. Perhaps most intriguingly, rodents do not have P2Y_11_ receptors [61,91]. Side-by-side comparisons of human and mouse macrophages might therefore serve to elucidate the advantages of P2Y_11_ expression, and possibly of dual G protein coupling. Similar to P2Y_6_ receptors [76], P2Y_11_ gene and protein expression increases in response to M-CSF [25,93], along with the scavenger receptor CD163, which is an M-CSF target gene and marker of M2 macrophages [100]. It is possible that P2Y_11_ expression in GM-CSF differentiated human monocytes is actually driven by M-CSF, because GM-CSF induces M-CSF production in monocytes [101]. By blocking TNF-α secretion and by promoting sTNFR2 release, P2Y_11_ may act as a sentinel in the surveillance of TNF-α driven inflammation [97]. P2Y_11_ thus represents a target for anti-inflammatory strategies in the treatment of autoimmune or infectious diseases. Agonistic P2Y_11_ receptor targeting to elicit anti-inflammatory P2Y_11_ signaling in macrophages might be desirable in the treatment of severe forms of coronavirus disease 2019 (Covid-19) that are characterized by excessive macrophage activation and TNF-α driven hyperinflammation [2,102]. Conversely, P2Y_11_ activation should be avoided in cancer treatments to prevent anti-inflammatory P2Y_11_ signaling that would otherwise limit therapy effectiveness.

## 7. P2Y_12_ Receptor

The G_i_-coupled, ADP-activated P2Y_12_ receptor is a well-known key player in platelet activation via crosstalk with the P2Y_1_ receptor in ADP-evoked intracellular Ca^2+^ responses [11,103]. While Gα subunits of the G_i_ protein inhibit AC to decrease cAMP-mediated PKA activation, Gβγ subunits activate phosphoinositide 3-kinase (PI3K) [104]. The first arm of P2Y_12_ receptor signaling is inhibited by pertussis toxin, which catalyzes the ADP-ribosylation of the Gα subunits of the heterotrimeric G_i_ protein. ADP-ribosylated Gα fails to inhibit adenylate cyclase activity, leading to an increase of intracellular cAMP levels. G_i_-coupled P2Y_12_ receptor signaling may also activate AKT (protein kinase B, PKB) and ERK [105]. G_i_-independent P2Y_12_ receptor signaling requires RhoA and Rho kinase [104] and may lead to p38 MAPK activation [106]. Moreover, Go coupling of P2Y_12_ receptor may also occur in some cell types [104].

P2Y_12_ receptor mRNA and protein expression has been detected in human monocytes and macrophages as well as in the human monocytic cell line THP1 [107,108]. In animal models, the P2Y_12_ receptor is expressed by hepatic macrophages, both from healthy, cirrhotic and cancerous liver [109]. P2Y_12_ receptor mRNA could be detected on human macrophages generated from monocytes with M-CSF/dexamethasone/IL-4 (MDI). Indeed, P2Y_12_ receptor mRNA was among the most significantly upregulated transcripts during MDI-induced differentiation. P2Y_12_ receptor protein expression has also been verified. Such macrophages display a phenotype similar to that of tumor-associated macrophages (TAM) [110]. In the same work, P2Y_12_ expression was confirmed on CD163-expressing TAM of melanoma in situ. P2Y_12_ expression has been reported to be robust in resting microglia, which are the primary immune sentinels of the central nervous system, but dramatically reduced after microglial activation [111]. In addition, FACS-purified peritoneal mouse macrophages have been shown to express P2Y_12_ receptors both at the mRNA and at the protein level [112]. Other immune cells, including dendritic cells, may also express P2Y_12_ receptors [113].

ADP-evoked intracellular Ca^2+^ responses in human monocytic THP-1 cells depended on both P2Y_12_ and P2Y_6_ receptors. P2Y_12_ receptor activation supported P2Y_6_ receptor-mediated intracellular Ca^2+^ signaling through suppression of AC activity in human monocytic cells [107]. In macrophage polarization (Figure 1), P2Y_12_ receptor expression is increased in alternatively activated M2 macrophages [114]. Pharmacological inhibition of P2Y_12_ receptor was found to evoke an endoplasmic reticulum (ER) stress response that appears to block M2 markers such as arginase-1, indicative of a shift towards M1 [109]. Similar to the P2Y_2_ receptor on macrophages, which senses ATP and UTP released from apoptotic cells [55], the P2Y_12_ receptor has been reported to direct migration of TAM towards ADP-releasing melanoma cells in necrotic tumor areas [110]. In recombinant U937 cells, stimulation of ectopic P2Y_12_ receptor with ADP induced the secretion of chemokines including CXCL8 [110].

Initially, P2Y_12_ receptor has been considered a microglia-specific receptor [111]. Later studies, however, demonstrated that it may also be expressed by peripheral macrophages such as peritoneal macrophages [112]. Nevertheless, the P2Y_12_ receptor is indeed enriched in microglia and therefore serves as a useful marker for the identification of human microglia, and P2Y_12_ receptor expression may help to discriminate activated microglia from quiescent microglia in the human CNS [115]. ADP-evoked calcium signaling in primary microglia is preferentially mediated by P2Y_12_ receptors. Upon traumatic brain injury, which is associated with lytic and controlled release of nucleotides (ATP/ADP), microglia are rapidly activated and extend cellular protrusions towards the site of injury [116], a process now known to be steered by P2Y_12_ receptors [111,117]. Consistently, microglia in P2Y_12_ receptor knockout mice showed significantly delayed responses toward sites of damage or local nucleotide injection in vivo [111]. P2Y_12_ receptor mediated integrin-ß1 activation is involved in the directional extension of microglia protrusions in brain tissue [118]. In the first step, when microglia activation is limited and occurs in the absence of cell body movement, microglia use their protrusions to seal small disruptions at the barrier between healthy and injured tissue. The subsequent downregulation of P2Y_12_ receptors causes the retraction of microglial protrusions. During severe traumatic brain injury, microglia undergo strong activation and give up residency. They acquire an amoeboid morphology, which is prerequisite to cell body movement, and migrate under the control of P2Y_12_ and P2X_4_ receptors [119]. PI3K, PLC, and Akt have been implicated in P2Y_12_ receptor-mediated microglia activation [120].

In addition to its cytoprotective role, P2Y12 receptors may contribute to the development of neuropathic pain during nerve injury induced microglia activation, and the stress MAPK p38 has been shown to be involved in the P2Y_12_-driven pathogenic process [106]. Likewise, P2Y_12_ receptors have been implicated in the generation of neuronal damage during brain ischemia [121].

In addition to the crucial role of P2Y_12_ receptor in macrophages, P2Y_12_ is also a major ADP receptor in osteoclasts [35]. While earlier work has implicated P2Y_1_, P2Y_2_, and P2Y_6_ receptors in the regulation of osteoclast function and survival [34,88,122], P2Y_12_ receptor has later been shown to be the primary ADP receptor in osteoclasts, suggesting that P2RY_12_ inhibition is a potential therapeutic target for pathologic bone loss [35].

Collectively, P2Y_12_ receptor is certainly abundant in macrophages and is a major or even the primary ADP receptor in microglia. P2Y_12_ receptor signaling may be anti-inflammatory to support the function of M2-like macrophages during tissue repair. P2Y_12_ receptor governs the migratory process of macrophages during these cytoprotective responses. However, P2Y_12_ receptor signaling may also contribute to the pathogenesis of certain diseases.

## 8. P2Y_13_ Receptor

Like the P2Y_12_ receptor, P2Y_13_ is a G_i_-coupled receptor activated by ADP [11]. Although it mainly inhibits AC-mediated cAMP production, it has also been reported to increase cAMP production in response to high agonist concentration [123]. In distinct cell types, P2Y_13_ receptor signaling depends on RhoA-mediated activation of ROCK [124,125].

P2Y_13_ receptor mRNA and protein is expressed in GM-CSF differentiated monocyte-derived macrophages [126]. The levels of P2Y_13_ receptor mRNA are similar in GM-CSF and M-CSF differentiated macrophages [25]. When P2Y receptor mRNA expression was analyzed in M-CSF differentiated monocyte-derived macrophages from ankylosing spondylitis patients, P2Y_13_ and P2Y_6_ receptor displayed the highest expression levels in macrophages with no significant differences between patients and healthy controls [127]. Human alveolar macrophages also express mRNA encoding P2Y_13_ receptor [26].

In human monocyte-derived macrophages, P2Y_13_ receptor supports P2X_4_ receptor-mediated Ca^2+^ signaling [126]. While P2Y_13_ along with the P2Y_11_ receptor was required for the amplitude of the nucleotide-evoked Ca^2+^ response, P2X_4_ receptor activity determined its duration. The P2Y_13_ receptor has been implicated in the defense against bacterial infection, and its expression was found to be increased in macrophages after LPS treatment as well as in tuberculosis patients in vivo [30]. P2Y_12_ and P2Y_13_ receptors were responsible for the ADP-mediated protection against bacterial infection, while P2Y_1_ was found to be dispensable. In response to ADP, P2Y_12_ and P2Y_13_ receptors mediated macrophage recruitment to the infected tissue most likely by stimulating the release of CCL2. ADP-induced P2Y_12_/P2Y_13_ receptor signaling required ERK activation and was counteracted by the elevation of cAMP.

P2Y_13_ receptor has also been implicated in cholesterol metabolism. In cultured hepatocytes, P2Y_13_ receptor activation has been shown to be essential for uptake of cholesterol-containing high-density lipoproteins (HDL) [128]. In macrophages, the P2Y_13_ receptor may play a similar role [129]. Loading of macrophages with cholesterol resulted in transcriptional reprogramming [130], characterized by the downregulation of P2Y_13_ and P2Y_14_ receptor mRNA. P2Y_13_ receptor mRNA was among the most significantly altered transcripts in response to cholesterol loading experiments. Downregulation of P2Y_13_ and P2Y_14_ receptor gene expression in response to excessive cholesterol loading appears to occur as part of a feedback regulation, preventing cholesterol overload in macrophages. Such a regulation may be particularly important for arterial macrophages, because dysregulated cholesterol metabolism is obviously pathogenic in atherosclerosis [131]. In addition to being an excess product intended for disposal, cholesterol and its metabolites can influence macrophage induction of inflammatory responses. Cholesterol may for instance regulate inflammasome activity or orchestrate antiviral responses of macrophages [131,132]. By controlling cholesterol uptake, the P2Y_13_ receptor may therefore play an important role in the regulation of macrophage inflammation.

Together with the chemokine receptors CCR2 and CCR4, P2Y_12_ and P2Y_13_ receptor were found to be part of a four-gene signature defining the tumor microenvironment of lung adenocarcinoma [133]. The infiltration of the tumor microenvironment by immune cells, including macrophages, which was controlled by these four genes, had predictive prognostic value in lung adenocarcinoma.

One of the main causes of osteoporosis in older women is the lack of estrogen. Although the P2Y_12_ receptor has been postulated to be the primary ADP receptor in osteoclasts [35], P2Y_13_ receptor knockout mice were found to be protected from ovariectomy-induced bone loss, indicating that targeted inhibition of P2Y_13_ receptor activity would also be desirable in estrogen deficiency-induced osteoporosis [125].

## 9. P2Y_14_ Receptor

The G_i_-coupled P2Y_14_ receptor (also known as GPR105) is activated by UDP as well as UDP-linked sugars, including UDP-glucose and UDP-galactose [11,39,40]. In addition to G_i_-mediated suppression of cAMP, P2Y_14_ signaling pathways may engage PI3-kinase-γ, GPCR kinases 2 and 3, PLC, and MAPK [134,135,136]. Moreover, P2Y_14_ signaling has recently been shown to stimulate expression and phosphorylation of STAT1 [136]. In human and mouse mucosal tissue, P2Y_14_ associated with epithelial cells can promote the production of chemokines such as CXCL8 [137]. Likewise, airway epithelial cells secrete CXCL8 in response to P2Y_14_ activation [138]. In contrast, in glioma cells P2Y_14_ receptor activation has been shown to decrease the expression of IL-6 [139], which may depend on cAMP [140]. Among immune cells, functional P2Y_14_ receptor expression has been reported for murine T cells [141], human and mouse neutrophils [142,143], as well as for a rat basophilic leukemia cell line [135].

Regarding macrophages, P2Y_14_ receptor mRNA expression has been detected in human alveolar macrophages [26] as well as in M-CSF differentiated monocyte-derived macrophages from healthy individuals and ankylosing spondylitis patients [127]. Murine bone marrow-derived macrophages have also been shown to express P2Y_14_ receptors [144].

Macrophage activation with LPS causes metabolic reprogramming, characterized by a switch from oxidative phosphorylation (OXPHOS) to glycolysis [145,146]. In addition, macrophages activate the concomitant synthesis and lysis of glycogen. The purpose of such an apparently futile metabolism is to accumulate glucose-6-phosphate for the generation of NADPH, the reduced form of nicotinamide adenine dinucleotide phosphate (NADP^+^) in the pentose phosphate pathway (PPP) [136]. Macrophages depend on NADPH to maintain high levels of glutathione, an endogenous antioxidant ensuring the survival of inflammatory macrophages.

UDP-glucose, a P2Y_14_ receptor agonist, is the activated form of glucose serving as the glucose donor in glycogen biosynthesis. Intriguingly, UDP-glucose produced during glycogenesis activates the P2Y_14_ receptor, resulting in the upregulation of the expression and phosphorylation of STAT1 [136], which is a key inflammatory transcription factor (Figure 3).

Enhanced glycogenesis thus facilitates macrophage inflammation via P2Y_14_ receptor-induced STAT1 activation and ensures macrophage survival by providing anti-oxidative NADPH.

Glycolysis can be inhibited by 2-deoxyglucose (2DG) at the level of hexokinase. Administration of 2DG during LPS-induced macrophage activation caused upregulation of P2Y_14_ receptor mRNA expression [147]. Although the meaning of P2Y_14_ receptor upregulation is currently unclear, it is conceivable that P2Y_14_ receptor-like P2Y_13_ receptor-supports the uptake of HDL [130]. Increased lipid availability may be required to fuel fatty acid oxidation and ATP production via OXPHOS, when glycolysis is inhibited [148].

In gout, a form of inflammatory arthritis, monosodium urate (MSU) crystals are deposited in joints. MSU can upregulate expression of P2Y_14_ receptor expression [149]. P2Y_14_ receptor suppresses cAMP, which is a negative regulator of the NLRP3 inflammasome [150]. P2Y_14_ thus facilitates inflammasome activation and caspase-1-mediated pyroptosis of macrophages, an inflammatory form of macrophage cell death known to contribute to the pathogenesis of gout. Finally, P2Y_14_ receptor may play a role in bone resorption, since P2Y_14_ receptor mRNA and protein were selectively upregulated during RANKL-induced osteoclast generation [151].

## 10. Concluding Remarks

The P2Y family of G protein-coupled receptors consists of eight members, which are classified in two subfamilies (Table 1). Further additions to the P2Y family are not expected. Due to the limited availability of suitable antibodies, P2Y receptor protein expression has not been examined systematically. Particularly, a comparative analysis of the expression of individual P2Y receptors at the protein level has not yet been performed in macrophages. Likewise, comprehensive data on P2Y receptor protein regulation during macrophage activation are missing. According to the current view, P2Y_1_ and P2Y_4_ receptors appear to play minor roles, while the functions of P2Y_2_, P2Y_6_, P2Y_11_, P2Y_12_, P2Y_13_ and P2Y_14_ are more established in macrophage biology. The P2Y_12_ receptor is strongly associated with microglia, the macrophages of the brain. Progress in P2Y receptor research has also been hampered, because selective agonists and antagonists are not yet available in the same manner for all P2Y receptor subtypes. The view that P2Y receptor agonists are generally pro-inflammatory still prevails [11] and has led to the conclusion that antagonists may be used to treat inflammatory conditions. The recent finding that P2Y_14_ receptors support inflammatory gene expression during TLR-induced macrophage activation confirms this view [136]. However, this concept has been challenged by a recent study, demonstrating that anti-inflammatory P2Y_11_ receptor signaling controls TNF-α driven inflammation [97]. P2Y_11_ is in many ways an unconventional member of the P2Y family of G protein-coupled receptors. Aside from being the only P2Y receptor coupling to both, G_q_ and G_s_ proteins, P2Y_11_ does not occur in rodents. This means that other models have to be established in order to promote P2Y_11_ translational research. Currently, only P2Y_2_ and P2Y_12_ receptors serve as clinical targets in dry eye and thrombotic disease, respectively. The other P2Y receptor subtypes lag behind in clinical development, although they may have great potential.

## Figures and Tables

**Figure 1 cells-10-01098-f001:**
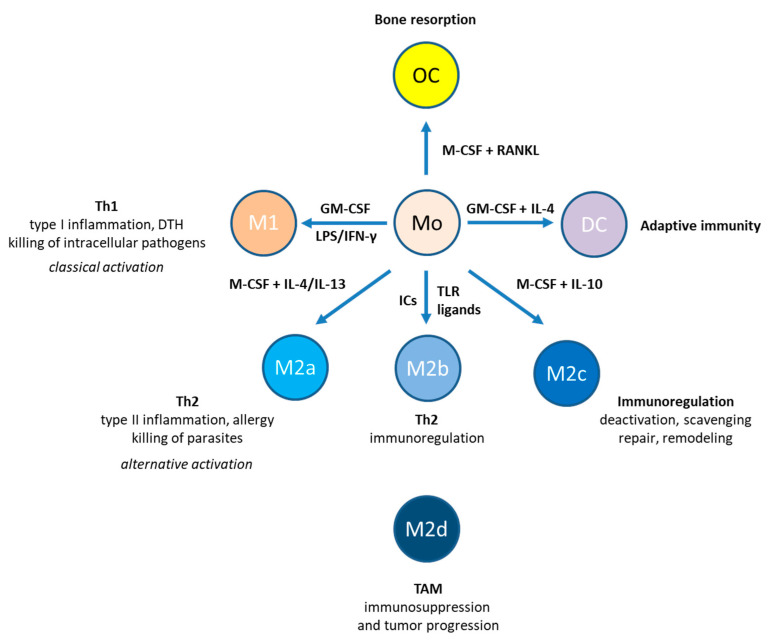
Monocyte-derived cell types. Monocytes can give rise to macrophages, osteoclasts (OC), and dendritic cells (DC). M1 macrophages are highly inflammatory and produce Th1-type cytokines. M2 macrophages can be divided into four subpopulations: M2a, M2b, M2c, and M2d. While M2a macrophages promote Th2-type inflammation, M2b macrophages perform immunoregulation. M2c macrophages completely deactivate inflammatory and immune responses. Tumor-associated macrophages are sometimes referred to as M2d macrophages, which have immunosuppressive, proangiogenic, and thus tumor-promoting properties. Abbreviations: DTH, delayed-type hypersensitivity; GM-CSF, granulocyte/macrophage colony-stimulating factor; IC, immune complex; IFN, interferon; IL, interleukin; LPS, lipopolysaccharide; M-CSF, macrophage colony-stimulating factor; RANKL, receptor activator of nuclear factor kappa-Β ligand; TAM, tumor-associated macrophages; Th, T helper; TLR, Toll-like receptor.

**Figure 2 cells-10-01098-f002:**
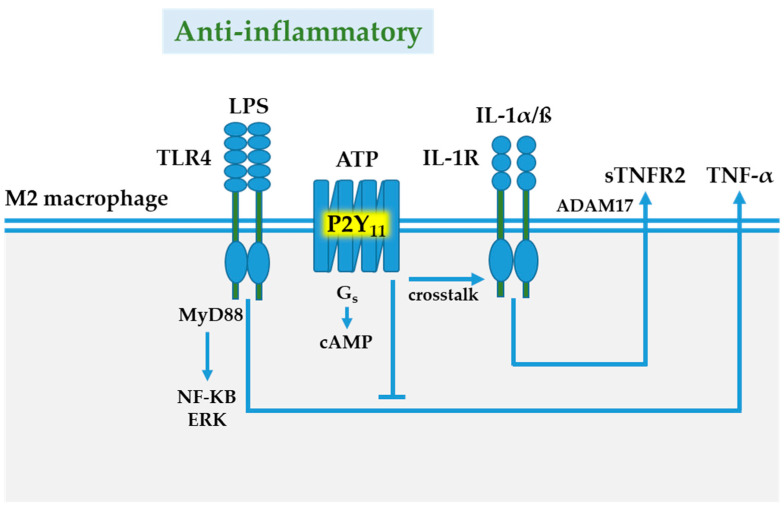
P2Y_11_ receptor-mediated anti-inflammatory responses. TLR4/IL-1R signaling via MyD88/NF-KB/ERK is a major pathway of macrophage activation. In anti-inflammatory M2 macrophages, the ATP-activated P2Y_11_ receptor can couple to G_s_ proteins causing an increase in intracellular cyclic AMP (cAMP), which on the one hand blocks TLR4-induced TNF-α secretion, and on the other, promotes the release of soluble tumor necrosis factor TNF receptor 2 (sTNFR2) via IL-1 receptor-mediated activation of metalloprotease ADAM17, also known as tumor necrosis factor-α converting enzyme (TACE). Abbreviations: ATP, adenosine triphosphate; extracellular signal-regulated kinase (ERK); IL, interleukin; LPS, lipopolysaccharide; nuclear factor kappa-Β; MyD88 (myleoid differentiation primary-response protein 88); TLR, Toll-like receptor.

**Figure 3 cells-10-01098-f003:**
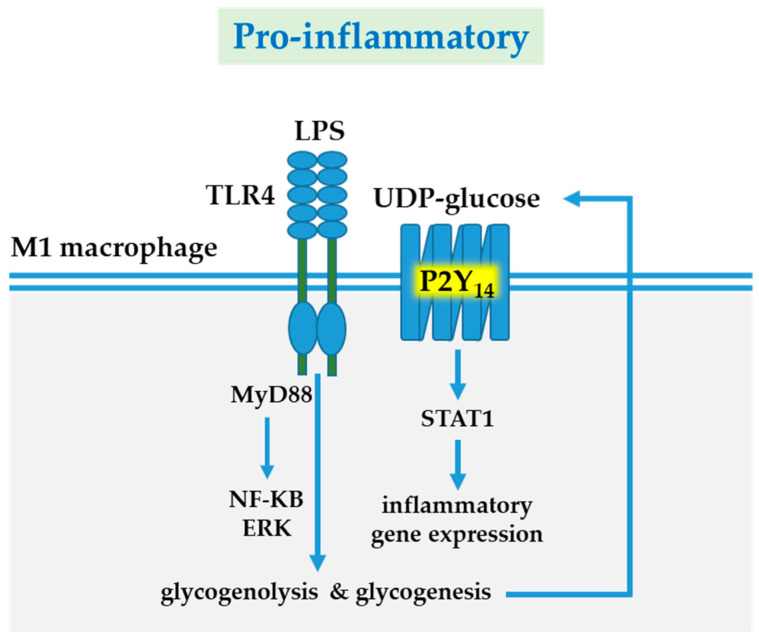
P2Y_14_ receptor-mediated pro-inflammatory responses. LPS-induced TLR4 signaling via MyD88/NF-KB/ERK is a major pathway of macrophage activation. In inflammatory M1 macrophages, LPS/IFNγ-enhanced glycogen metabolism (synthesis and lysis) causes accumulation and release of UDP-glucose, resulting in P2Y_14_ receptor activation. P2Y_14_ receptor signaling upregulates expression and phosphorylation of STAT1. Abbreviations: extracellular signal-regulated kinase (ERK); IFN, interferon; LPS, lipopolysaccharide; nuclear factor kappa-Β; MyD88 (myleoid differentiation primary-response protein 88); STAT1, signal transducer and activator of transcription 1; TLR, Toll-like receptor; UDP, uridine diphosphate.

**Table 1 cells-10-01098-t001:** Human P2Y receptors.

P2Y Group	Official Gene Symbol	Protein Name	Agonist(s)	G Protein Coupling
P2Y_1_-like	P2RY1	P2Y_1_	ADP	G_q_
	P2RY2	P2Y_2_	ATP ≈ UTP	G_q_-G_i_
	P2RY4	P2Y_4_	UTP	G_q_-G_i_
	P2RY6	P2Y_6_	UDP	G_q_
	P2RY11	P2Y_11_	ATP	G_q_-G_s_
P2Y_12_-like	P2RY12	P2Y_12_	ADP	G_i_
	P2RY13	P2Y_13_	UDP-glucose	G_i_
	P2RY14	P2Y_14_	UDP	G_i_

## Data Availability

Not applicable.

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
