# Peer review of "Control of Macrophage Inflammation by P2Y Purinergic Receptors"

_cells, 2021, doi:10.3390/cells10051098_

Round 1
Reviewer 1 Report
Klaver and Thurnher have compiled a well-written review of T2Y purinergic signaling in macrophages. The review is detailed and organized. I have a few suggestions that may improve the present manuscript.
Please incorporate referencing Figure 2, Figure 3, and the Table in the body of the text.
Is there evidence for P2Y1 forming a heterodimer with either P2Y11 or P2Y12? If so, please cite, lines 152-154.
It is unclear what “priming with nucleotides” means, line 206.
Please be more specific with the term “Nanoparticles,” line 213. Nanoparticles come in many different flavors and have different affinities for different cells types.
Lines 222-223 indicate that P2Y11, is absent in mice. Please provide a brief discussion on the implications for both translational and mouse research.
Lines 241-243 state that P2Y2 activation may initiate the coagulation cascade, please provide a citation.
The P2Y6 section begins stating that P2Y6 is preferentially activated by UDP, however line 303 states that UDP is a poor agonist. Please clarify. If this is a species-specific difference between mouse and human, please discuss.
Line 443 makes a claim that targeting P2Y11 signaling may be desirable for the treatment of COVID-19. To me, this is a bold claim that may not be appropriate to include in the present review, particularly as the pathophysiology of COVID-19-associated respiratory disease is still poorly understood.
A vast majority of the evidence presented describing P2Y12 activation supports its role in promoting an M2-like phenotype, but at the very end of this section, there is a statement that it may be pro-inflammatory through activation of the inflammasome. Either provide this additional evidence or remove this sentence, lines 522-524. Presently, it is confusing to the reader
The manuscript ends rather abruptly. I would suggest including either a summary or concluding paragraph.
Author Response
Thank you very much for the careful consideration of our manuscript and the helpful comments. We have addressed all your points:
Please incorporate referencing Figure 2, Figure 3, and the Table in the body of the text.
Reply: We refer to Figure 1 on page 1 line 43/44, to Figure 2 on page 9 line 418 and to Figure 3 on page 13 line 603/604. We refer to Table 1 on page 3 line 76.
Is there evidence for P2Y1 forming a heterodimer with either P2Y11 or P2Y12? If so, please cite, lines 152-154.
Reply: We have added a reference for P2Y11. We removed P2Y12 because there was no appropriate PubMed-listed work to cite, although I remember that the Kennedy lab (?) reported such heterodimers some years ago.
It is unclear what “priming with nucleotides” means, line 206.
Reply: In this case “nucleotides” refers to ATP and UTP. The priming procedure is explained in lines 202/203.
Please be more specific with the term “Nanoparticles,” line 213. Nanoparticles come in many different flavors and have different affinities for different cells types.
Reply: We have specified the nanoparticles used in that study.
Lines 222-223 indicate that P2Y11, is absent in mice. Please provide a brief discussion on the implications for both translational and mouse research.
Reply: In addition to our statement in lines 434-436, we address this point in the new “Concluding remarks”.
Lines 241-243 state that P2Y2 activation may initiate the coagulation cascade, please provide a citation.
Reply: We have added the appropriate reference.
The P2Y6 section begins stating that P2Y6 is preferentially activated by UDP, however line 303 states that UDP is a poor agonist. Please clarify. If this is a species-specific difference between mouse and human, please discuss.
Reply: The wording is indeed misleading. In that work (del Rey J Biol Chem 2006), resting macrophages showed weak responses to UDP (purified by hexokinase to remove ATP), however, not because UDP is a poor agonist but rather because of low P2Y6 receptor expression. Upon macrophage activation, UDP responsiveness (i.e. P2Y6 receptor expression) increased. Of note, in many studies P2Y receptor detection at the protein level has often been impossible due to the lack of appropriate antibodies.
Line 443 makes a claim that targeting P2Y11 signaling may be desirable for the treatment of COVID-19. To me, this is a bold claim that may not be appropriate to include in the present review, particularly as the pathophysiology of COVID-19-associated respiratory disease is still poorly understood.
Reply: TNF-α is an important component of hyperinflammation associated with Covid-19 and P2Y11 receptor has been shown to serve as a sentinel of TNF-α induced inflammation. We therefore believe that it is warranted to suggest P2Y11 as a potential target in the treatment of hyperinflammation. However to address the reviewer’s point, we tempered our conclusion and added a reference to emphasize the role of TNF-α in Covid-19.
A vast majority of the evidence presented describing P2Y12 activation supports its role in promoting an M2-like phenotype, but at the very end of this section, there is a statement that it may be pro-inflammatory through activation of the inflammasome. Either provide this additional evidence or remove this sentence, lines 522-524. Presently, it is confusing to the reader
Reply: The sentence has been re-written: “However, P2Y12 receptor signaling may also contribute to the pathogenesis of certain diseases.” We describe the pathogenic role of P2Y12 in lines 510-519.
The manuscript ends rather abruptly. I would suggest including either a summary or concluding paragraph.
Reply: We have added a concluding paragraph.
Reviewer 2 Report
The paper highlights an interesting topic, however not very innovative.
The manuscript is written in a good style and easy to read, there are no grammatical, punctuation or linguistic errors. In the manuscript there are good, clear figures and tables that facilitate understanding of the issues described.
In my opinion, a few summary sentences are missing at the end of the manuscript.
The paper is valuable to publish in Cells.
Author Response
Thank you for reviewing our manuscript. We have addressed your point:
In my opinion, a few summary sentences are missing at the end of the manuscript.
Reply: We have added a concluding paragraph.
Reviewer 3 Report
The manuscript entitled ´control of macrophage inflammation by P2X purinergic receptors´ reveals an excellent review on the important function of the P2X receptor Family on immunological function of macrophages. The authors provide a detailled and fundamental literature review. I can strongly recommend it for publication.
Author Response
Thank you for reviewing our manuscript.